# Profiling and Functional Analysis of mRNAs during Skeletal Muscle Differentiation in Goats

**DOI:** 10.3390/ani12081048

**Published:** 2022-04-18

**Authors:** Siyuan Zhan, Hongfan Zhai, Min Tang, Yanan Xue, Dandan Li, Linjie Wang, Tao Zhong, Dinghui Dai, Jiaxue Cao, Jiazhong Guo, Li Li, Hongping Zhang

**Affiliations:** Farm Animal Genetic Resources Exploration and Innovation Key Laboratory of Sichuan Province, College of Animal Science and Technology, Sichuan Agricultural University, Chengdu 611130, China; siyuanzhan@sicau.edu.cn (S.Z.); hongfanz@163.com (H.Z.); tangminsicau@163.com (M.T.); xyn19960317@163.com (Y.X.); lidandan@sicau.edu.cn (D.L.); wanglinjie@sicau.edu.cn (L.W.); zhongtao@sicau.edu.cn (T.Z.); 71317@sicau.edu.cn (D.D.); jiaxuecao@sicau.edu.cn (J.C.); jiazhong.guo@sicau.edu.cn (J.G.); lily@sicau.edu.cn (L.L.)

**Keywords:** muscle differentiation, transcriptome analysis, mRNAs, goat

## Abstract

**Simple Summary:**

The growth and development of skeletal muscle is strictly regulated by complex gene networks. In order to explore the genes involved in regulating the differentiation of skeletal muscle satellite cells (MuSCs) in goats, RNA sequencing was used to characterize gene expression profiles during the MuSC differentiation, and to identify differentially expressed genes (DEGs) among the different stages. A total of 2551 DEGs were found. Functional enrichment analysis revealed genes involved in muscle development-related GO terms and pathways, such as muscle structure development, muscle contraction, muscle cell development, muscle cell differentiation, and the MAPK signaling pathway. This study will be useful for future studies on muscle growth and development in goats.

**Abstract:**

Skeletal myogenesis is a complicated biological event that involves a succession of tightly controlled gene expressions. In order to identify novel regulators of this process, we performed mRNA-Seq studies of goat skeletal muscle satellite cells (MuSCs) cultured under proliferation (GM) and differentiation (DM1/DM5) conditions. A total of 19,871 goat genes were expressed during these stages, 198 of which represented novel transcripts. Notably, in pairwise comparisons at the different stages, 2551 differentially expressed genes (DEGs) were identified (*p* < 0.05), including 1560 in GM vs. DM1, 1597 in GM vs. DM5, and 959 in DM1 vs. DM5 DEGs. The time-series expression profile analysis clustered the DEGs into eight gene groups, three of which had significantly upregulated and downregulated patterns (*p* < 0.05). Functional enrichment analysis showed that DEGs were enriched for essential biological processes such as muscle structure development, muscle contraction, muscle cell development, striated muscle cell differentiation, and myofibril assembly, and were involved in pathways such as the MAPK, Wnt and PPAR signaling pathways. Moreover, the expression of eight DEGs (MYL2, DES, MYOG, FAP, PLK2, ADAM, WWC1, and PRDX1) was validated. These findings offer novel insights into the transcriptional regulation of skeletal myogenesis in goats.

## 1. Introduction

Skeletal muscle tissue accounts for approximately 40–60% of the entire body weight of adult ruminants, which not only serves as the determining factor for the performance of meat production but also has critical influences on the meat quality [1]. Skeletal muscle primarily comprises thousands of multinucleated contractile myofibers originating from myoblasts. Myogenesis refers to the formation of functional myofibers [2], and during the course of myogenesis, a population of myogenic cells become quiescent and are located surrounding muscle fibers in mature muscle, which are termed satellite cells [3]. The proliferation and fusion of satellite cells with the existed muscle fiber are vital for postnatal muscle fiber hypertrophy [4]. As a result, having a larger number of muscle fibers and satellite cells is beneficial for meat production purposes.

Myogenesis, a complex biological process, involves a variety of gene regulatory networks, including myogenic regulatory factors (MRFs), myocyte enhancer factor 2 (MEF2), paired box 3 (Pax3) and Pax7 [5]. The ordered and coordinated expression of these genes eventually promotes skeletal muscle growth. The MRF family is a class of transcription factors containing helix-loop-helix (BASIC helix-loop-helix) structures, which are mainly composed of four members: MyoD (Myogenic Differentiation), Myf5 (Myogenic Factor 5), Myf6 (Myogenic Factor 5) and MyoG (Myogenin) [6]. These four genes are critical in early skeletal muscle development, in which they are sequentially expressed. In addition, the transforming growth factor β (TGF-β), insulin-like growth factor (IGF), and fibroblast growth factor (FGF) families have the most significance for muscle development [7]. In addition to these commonly known genes, the others that may have an impact onmyogenesis remain unidentified and/or uncharacterized. Many gene expression profiling studies in agriculturally significant animals have been carried out since the introduction of high-throughput sequencing tools, and many genes or factors associated to skeletal muscle development have been found [8,9,10,11]. However, in most studies, tissue samples have been used to explore critical regulators involved in muscle growth and development. Because tissue samples contain a variety of cell types that might affect identification accuracy, the use of cell samples to identify important regulators of myogenesis is a useful method.

Domestic goats (*Capra hircus*) are one of the world’s earliest domesticated livestock species, with their meat, milk, hair, and skins being used all over the world. Goat meat production is crucial for a variety of industries, particularly in the livestock business of developing countries. Nevertheless, the way in which goat muscle fibers are formed and the underlying mechanisms remain elusive. We proposed that key genes and signaling pathways that control myogenic differentiation in goats have yet to be identified and characterized. To this end, we analysed the transcriptomic changes in goat skeletal muscle satellite cells (MuSCs) cultured under proliferation and differentiation conditions, and identified several candidates. Our findings are expected to pave the way for future research into goat skeletal muscle development.

## 2. Materials and Methods

### 2.1. MuSC Culture and Differentiation

The Animal Care and Use Committee of the College of Animal Science and Technology, Sichuan Agricultural University, Sichuan, China, approved all of the animal care, slaughter, and experimental procedures in accordance with the Regulations for the Administration of Affairs Concerning Experimental Animals (Ministry of Science and Technology, China) [SAU201418]. Primary MuSCs were isolated and cultured from fetal goat (Chengdu Ma goat, female, *n* = 1)-derived *longissimus* muscle, as previously described [12,13]. The MuSCs were seeded in 6-well plates (at a density of ~2 × 10^4^ cells/well) and cultured in Dulbecco’s Modified Eagle Medium in a 5% CO_2_ incubator at 37 °C. Next, 10% fetal bovine serum (Gibco, Invitrogen, Carlsbad, CA, USA) and 2% penicillin and streptomycin (Invitrogen) solution were supplemented to the growth medium (GM). When the MuSCs reached 80–90% confluence, the growth medium was replaced with differentiation medium (DM) containing DMEM, 2% horse serum (Gibco) and 2% penicillin and streptomycin in order to induce MuSC myogenic differentiation. The medium was replaced with fresh medium every 48 h. Proliferating MuSCs were labelled GM samples, while MuSCs differentiated for 1 and 5 days were labelled as DM1 and DM5 samples (biological triplicates were included for all of the conditions). All of the samples were kept at −80 °C before RNA extraction.

### 2.2. RNA Extraction and Sequencing

The total RNA was extracted using a TRIzol reagent kit according to the manufacturer’s instructions (Invitrogen, Carlsbad, CA, USA). The RNA quality was assessed on an Agilent 2100 Bioanalyzer (Agilent Technologies, Santa Clara, CA, USA) and checked using RNase-free agarose gel electrophoresis. After removing the rRNAs, the enriched RNAs were fragmented into short fragments and reverse-transcribed into cDNAs. Next, the cDNA fragments were purified with a Qiagen Quick PCR extraction kit (Qiagen, Hilden, Germany), end repaired, base added, and ligated to the sequencing adapters. Then, uracil-N-glycosylase (UNG) was used to digest the second-strand cDNA. The digested products were size-selected by gel electrophoresis before PCR amplification, and were sequenced using an Illumina HiSeqTM 4000 by Gene Denovo Biotechnology Co. (Guangzhou, China).

### 2.3. Differentially Expressed Gene Analysis

The fragments per kilobase of transcript per the million mapped reads (FPKM) value was calculated in order to quantify its expression abundance and variations using StringTie [14] software (v1.3.1). Differential expression analysis was performed using DESeq2 [15] software between two different groups. A false discovery rate (FDR) < 0.05 and |log2(Fold Change)| > 1 were used as cut-offs to calculate the differentially expressed genes.

### 2.4. Expression Trend Analysis

In order to examine the expression pattern of DEGs, the expression data of each sample were normalized to 0, log2(v1/v0), log2(v2/v0), and then clustered using Short Time-series Expression Miner software (STEM, v1.3.13) [16]. Then, analysis with these parameters showed that the maximum unit change in the model profiles between time points was 1, the maximum output profile number was 20 (similar profiles were merged), and the minimum ratio of fold change of DEGs was no less than 2.0. The clustered profiles with *p*-values ≤ 0.05 were considered statistically significant.

### 2.5. GO and KEGG Pathway Analysis

All of the DEGs were mapped to GO terms in the Gene Ontology database (http://www.geneontology.org/ (accessed on 5 January 2022)). Significantly enriched GO terms were defined by a hypergeometric test (FDR ≤ 0.05). Pathway enrichment analysis identified significantly enriched metabolic pathways or signal transduction pathways in the DEGs. The calculation formula was the same as that used in the GO analysis. After multiple test correction, pathways with a Q value less than 0.05 were defined as being significantly enriched.

### 2.6. Validation of the Sequencing Results by qRT-PCR

The qRT-PCR primers for DEGs and internal control genes were designed using Primer-BLAST (http://www.ncbi.nlm.nih.gov/tools/primer-blast/ (accessed on 10 November 2021)), and are listed in Table 1. For quantification, the GAPDH gene was employed as an internal standard. Each sample had 3 biological replicates. The PCR system (10 µL) consisted of 2 × M5 HiPer SYBR Permix EsTaq 5 µL, 0.4 µL of forward and reverse primers (10 μmol), 1 µL template cDNA, and ddH_2_O. The optimal reaction programs included denaturation at 95 °C for 30 s, followed by 39 cycles of 95 °C for 10 s and 30 s at the Tm indicated in Table 1. The melting curves were analysed between 65 and 95 °C with increments of 0.5 °C. The relative expression levels of the genes were calculated by the 2^−ΔΔCt^ method [17] and normalized to the controls.

### 2.7. Statistical Analysis

One-way analysis of variance (ANOVA) was performed by IBM SPSS Statistics 27.0.0. The least significant difference (LSD) method was used to test the difference significance. GraphPad Prism 8 (GraphPad, San Diego, CA, USA) was used to prepare the figures and diagrams. Comparisons with *p* values less than 0.05 were considered statistically significant, and those with *p* values less than 0.01 were considered extremely significant.

## 3. Results

### 3.1. MuSC Differentiation Program

Quiescent MuSCs were converted to myoblasts and allowed to proliferate in growth medium (GM) until they achieved 80% confluence (Figure 1A). This was deemed the proliferation phase. The myoblasts were then differentiated into myotubes in differentiation medium (DM). The cells differentiated to elongated myocytes one day after the replacement of GM with DM (Figure 1B). The myocytes then fused to form long, multinucleated myotubes on the fifth day (Figure 1C). Samples were collected during the proliferation phase (GM), and one day (DM1) and five days (DM5) after DM replacement for further transcriptomic analysis.

### 3.2. Temporally Regulated Transcription during Skeletal Myogenesis

In order to identify prospective genes that regulate skeletal myogenesis in goats, we performed mRNA-seq to characterize gene transcription profiles in MuSCs at different phases (GM, DM1 and DM5). We were able to recover over 77,705,900 clean reads (>99.72% of raw data) in each sample processed, and the uniquely-mapped reads were more than 89.61%. The GC content and read distribution from all of the samples were highly reliable and sufficient for downstream computational analysis. Specifically, the mean GC content was 50.29%, and the Q30s of all of the samples were >93.74% (Table 2). Our mRNA-seq results recovered 19,871 mRNA transcripts in total, 198 of which were novel and previously unidentified. Figure 2A summarizes the abundance of mRNA from these samples as fragments per kilobase per million (FPKM) values. Figure 2B depicts the expression of different samples. The complicated sample composition was represented by the two parameters on both the x- and y-axes, which helped to determine and visualize the variation between the samples. Remarkably, all of the samples were nicely separated into three sections coinciding with their phases, suggesting high reproducibility between biological replicates (Figure 2C). Next, we conducted clustering analysis based on sample similarities (Figure 2D). Our analysis again showed a reliable clustering between samples at each time point.

### 3.3. Differentially Expressed Genes (DEGs) Analysis

Based on our differential analysis, DEGs were defined as genes that were differentially expressed between stages using FDR < 0.05 and Fold change > 2 as cut-offs. Noticeably, we observed more upregulated DEGs than downregulated DEGs during MuSC differentiation (Figure 3A and Appendix A). A total of 12,756 differentially expressed genes were identified, and 1560 DEGs were identified when comparing the GM and DM1 groups, among which 813 showed increased expression and 747 showed decreased expression. Moreover, the number of DEGs detected from the GM and DM5 comparison groups was 1597, the number of upregulated genes was 1097, and the number of downregulated genes was 500. In addition, 959 DEGs were identified in the DM1 and DM5 comparison groups, and 722 DEGs were upregulated and 237 DEGs were downregulated. We next performed unsupervised clustering analysis according to the temporal gene expression profile in order to reveal the similarities between various temporal phases (Figure 3B–D). The Venn diagram in Figure 3E depicts the common and distinct DEGs expressed in different phases (Figure 3E). We observed 202 shared genes in the intersection, the expression levels of which are summarized in Appendix A.

### 3.4. Expression Trend Analysis

A total of 2551 DEGs were clustered according to the expression trend among the three stages. There were eight temporal expression patterns identified, of which three expression profiles were significantly enriched, including two upregulated patterns (profiles 7 and 6) and one downregulated pattern (profile 1). Specifically, profiles 7, 6 and 1 contained 468, 559, and 387 DEGs, respectively (Figure 4B). Therefore, 1027 upregulated DEGs and 387 downregulated DEGs were enriched in these three expression profiles. These results revealed the temporal gene expression changes in MuSCs during proliferation and differentiation, providing a reliable dataset for future candidate gene screening.

### 3.5. GO Enrichment Analysis

In order to gain further insights into how these DEGs regulated myogenesis, we took advantage of the OmicShare tools (http://omicshare.com/tools (accessed on 10 January 2022)). GO terms were assigned to a total of 2551 DEGs. We found that the DEGs from GM vs. DM1, GM vs. DM5, and DM1 vs. DM5 were significantly enriched for 640 GO terms (Q value < 0.05, 505 biological process GO terms, 80 cellular component GO terms, and 55 molecular function GO terms) (Appendix A), 825 GO terms (610 biological process GO terms, 128 cellular component GO terms, and 87 molecular function GO terms) (Appendix A), and 406 GO terms (322 biological process GO terms, 55 cellular component GO terms, and 29 molecular function GO terms), respectively (Appendix A). Among all of the differentially expressed genes, we focused on the comparison between DM1 and DM5. The top 20 enriched GO terms are shown in Figure 5. Intriguingly, we found several biological process terms related to muscle development, including the muscle system process, muscle structure development, muscle cell development, striated muscle contraction, and striated muscle cell development. In terms of the cellular component GO terms, the top hits were contractile fiber, the actin cytoskeleton, and contractile fiber. Within the molecular function category, the most enriched GO terms were related to protein binding and structural constituents of muscle. GM vs. DM1 and GM vs. DM5 were also analysed, and the results are shown in Appendix A.

### 3.6. KEGG Enrichment Analysis

We also performed KEGG enrichment analysis of the DEGs in order to further explore the signaling pathways through which DEGs could regulate muscle differentiation. For the three comparison groups GM vs. DM1, GM vs. DM5, and DM1 vs. DM5, the DEGs were enriched in 21, 41, and 21 pathways, respectively. Figure 6 summarizes the top 20 pathways identified in each comparison (Figure 6A–C). Interestingly, we were able to find many pathways that were previously known to be involved in myogenesis, including the Wnt, MAPK, Hippo, p53 and PPAR signaling pathways (Appendix A).

### 3.7. Validation of the DEGs

In order to validate the high-throughput RNA-seq results, we selected and examined the expression levels of eight DEGs (MYL2, DES, MYOG, FAP, PLK2, ADAM, WWC1, PRDX1) by qRT-PCR. The results showed that the expression patterns of these eight genes nicely replicated the expression trends calculated from the RNA-seq data, which confirmed the reliability of the sequencing data (Figure 7).

## 4. Discussion

Skeletal muscle development relies on myoblast differentiation and proliferation, and is a crucial factor influencing growth rate, meat quality and yield, as well as other important economic traits of livestock [18]. Myogenesis is tightly modulated, with multiple critical genes involved, such as the MRF (myogenic regulatory factor) [19,20] and MEF2 (myocyte enhancer factor-2) [21,22] families, MSTN (Myostatin) [23,24], and IGFs (Insulin-like growth factors) [25,26]. However, apart from these widely known genes, many other genes influencing myogenesis remain unidentified and/or uncharacterized. Although progress has been made in understanding how genes work during myogenesis in pigs [8,10], cattle [27], goats [11,28], and sheep [9], our understanding of the genes involved in goat skeletal muscle development is still limited and incomplete. Our work systematically characterized temporal gene expression during goat MuSC myogenic differentiation. We discovered that 2551 mRNAs were differentially expressed, and more importantly, 202 mRNAs were found to be differentially expressed across all of the comparisons, suggesting their critical roles in muscle differentiation. Similarly, Zhang et al. found 4820 differentially expressed genes during the proliferation and myogenic differentiation phases of buffalo MuSCs, and identified a number of functionally important genes [29]. Remarkably, all of the eight randomly selected DEGs (MYL2, DES, MYOG, FAP, PLK2, ADAM, WWC1, PRDX1) were validated using qRT-PCR, suggesting that our identification of DEGs through the high-throughput RNA-seq approach was highly reliable and reproducible. MYOG is a member of the muscle regulatory factor (MRF) family, and is involved mainly in the fusion and differentiation of myoblasts [5]. MYL2 belongs to the myosin light chain (MYL) family, and plays a vital role in muscle growth and contraction [30]. In addition, the ADAM (A Disintegrin And Metalloprotease) gene family encodes a diverse group of transmembrane cell-surface proteins with adhesion and proteolytic functions, and is involved in a broad range of biological processes, such as cell fusion, spermatogenesis, and development [31]. We used STEM software, which is commonly used to research dynamic biological processes [16,32], to identify relevant temporal expression profiles and the genes associated with these profiles because our data were obtained at multiple time points. Then, the most important profiles were saved and aggregated to form clusters for subsequent study [33]. We identified three significant expression profiles, including two upregulated patterns (profiles 7 and 6) and one downregulated pattern (profile 1). Moreover, 1027 upregulated DEGs and 387 downregulated DEGs were enriched in these expression profiles, suggesting that genes involved in myogenesis are dynamically regulated at differentiation stages.

We then performed enrichment analysis to further dive into the molecular mechanisms by which the DEGs could regulate myogenesis. In this study, DEGs from GM versus DM1, GM versus DM5, and DM1 versus DM5 were significantly enriched for 640 GO terms, 825 GO terms and 406 GO terms, respectively. Among them, we indeed identified many terms related to muscle development, such as muscle structure development, muscle tissue development, muscle contraction, muscle cell development, and muscle cell differentiation. These findings suggest that other candidate genes are likely to regulate MuSC proliferation and differentiation. For example, Muscle LIM protein (MLP), a member of the cysteine-rich protein (CRP) family that is produced primarily in skeletal muscle and myocardium, is encoded by cysteine and glycine rich protein 3 (CSRP3) [34,35]. MLP has been demonstrated to stimulate myogenic differentiation by binding to transcription factors such as MyoD and Myogenin in the nucleus [36]. In addition, myopalladin (MYPN) is an immunoglobulin-containing protein that is specifically expressed in striated muscle tissues. Subcellularly, MYPN is found in the Z−line and I−band of the sarcomere, as well as the nucleus. MYPN promotes skeletal muscle growth by activating the serum response factor (SRF) pathway in muscle [37]. Moreover, the ACTA1 and TNNT1 genes are involved in muscle fiber formation. [38]. Despite the limitations of enrichment analysis, these annotations provide useful hints and directions for the downstream characterization of the regulatory mechanisms involved in myogenesis.

It is unsurprising to see that genes involved in the same pathway are synergistically regulated. Pathway-based analysis can further assist us in understanding biological functions [39]. We therefore examined the possible roles of DEGs through pathway enrichment analysis. Fifty-one pathways stood out as being significantly enriched among the three comparison groups. Among them, we were vindicated to find many pathways that reportedly control muscle growth and development, including but not limited to the MAPK, Wnt, and PPAR signaling pathways and cell cycle regulators. The MAPK family plays crucial roles in complex cellular processes, such as proliferation, differentiation, and development, by regulating the cell cycle and other cell proliferation-associated proteins [40]. Previous studies have shown that the JNK/MAPK and P38/MAPK signaling pathways play essential roles in myogenesis [41]. In addition, a tightly regulated cell cycle is indispensable for muscle development. For example, the timing of muscle cell cycle entry and exit are known to be crucial for optimal muscle fiber structures. Meanwhile, the Wnt signaling pathway—with its active roles in embryonic myogenesis—was noticeable in our profile as well. A previous study on Wnt revealed that lacking several vital Wnt effectors could lead to substantial tissue damages, while poor muscle development could cause death in mice, indicating the critical role of the Wnt pathway in prenatal myogenic development [42]. Furthermore, the Wnt pathway also participates in the proliferation of signaling satellite cells in skeletal muscle during muscle cell regeneration [43,44]. Finally, the PPAR signaling pathway was also highlighted. It has been well established that both adipocytes and skeletal muscle cells derive from mesenchymal pluripotent cells [45,46]. During near mid-gestation, fetal skeletal muscle contains a great number of pluripotent cells, which differentiate into either adipogenic or myogenic cells [47,48]. Altogether, the pathways highlighted in the enrichment analysis underscore the significance of coordinated regulation in accurate and complete myogenic differentiation.

## 5. Conclusions

Our work not only phenotypically characterized the different phases underlying goat MuSC differentiation but also examined the temporal changes in transcriptional expression. We identified 202 that were differentially expressed across all of the comparisons between the proliferation and myogenic differentiation phases of MuSCs, which likely contain key and novel regulators. In addition, we further verified selected DEGs (MYL2, DES, MYOG, FAP, PLK2, ADAM, WWC1, and PRDX1). These findings offer novel insights into the transcriptional regulation of skeletal myogenesis, and provide a reference target for the further study of skeletal muscle development in goats.

## Figures and Tables

**Figure 1 animals-12-01048-f001:**
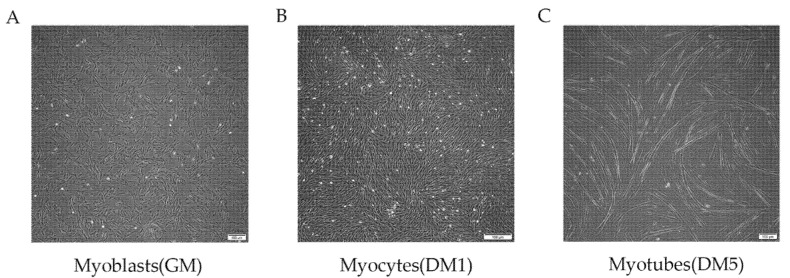
MuSC differentiation program; phase-contrast micrographs depicting cultures of goat MuSCs in proliferating conditions (growth medium (GM)) (**A**), in early myogenesis after 24 h in differentiation medium (Myocytes, DM1) (**B**), and in late myogenesis, including myotubes, after 5 days in DM (Myotubes, DM5) (**C**).

**Figure 2 animals-12-01048-f002:**
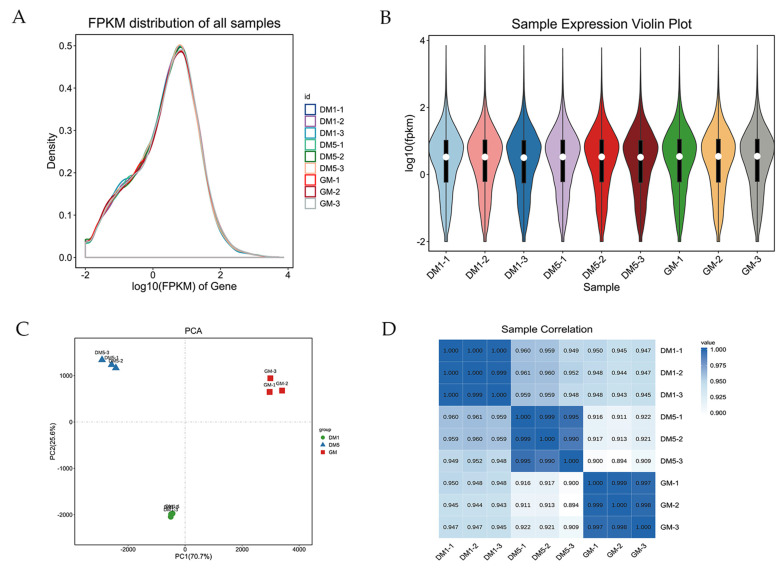
Expression analysis of the mRNA. (**A**) The density distribution of mRNAs was according to log10 (FPKM); (**B**) the nine sample expressions of GM−1, GM−2, GM−3, DM1−1, DM1−2, DM1−3, DM5−1, DM5−2, of DM5−3 in a violin plot, which was replaced by log10(FPKM); (**C**) the PCA analysis of 9 samples; (**D**) the sample relationship cluster analysis.

**Figure 3 animals-12-01048-f003:**
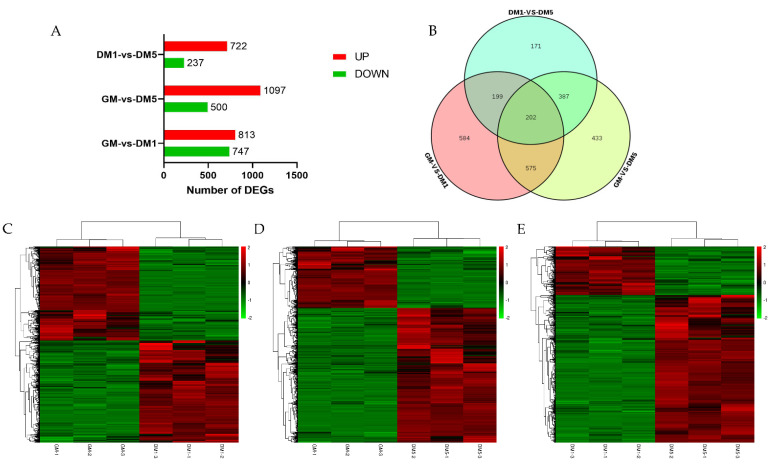
Differentially expressed genes (DEGs) analysis. (**A**) Numbers of upregulated and downregulated genes in goat MuSCs in three phases. (**B**) A venn diagram depicting the DEGs in various phases. (**C**–**E**) Hierarchical clustering analysis of DEGs through pairwise comparisons. Red denotes a high level of expression, whereas green denotes a low level of expression.

**Figure 4 animals-12-01048-f004:**
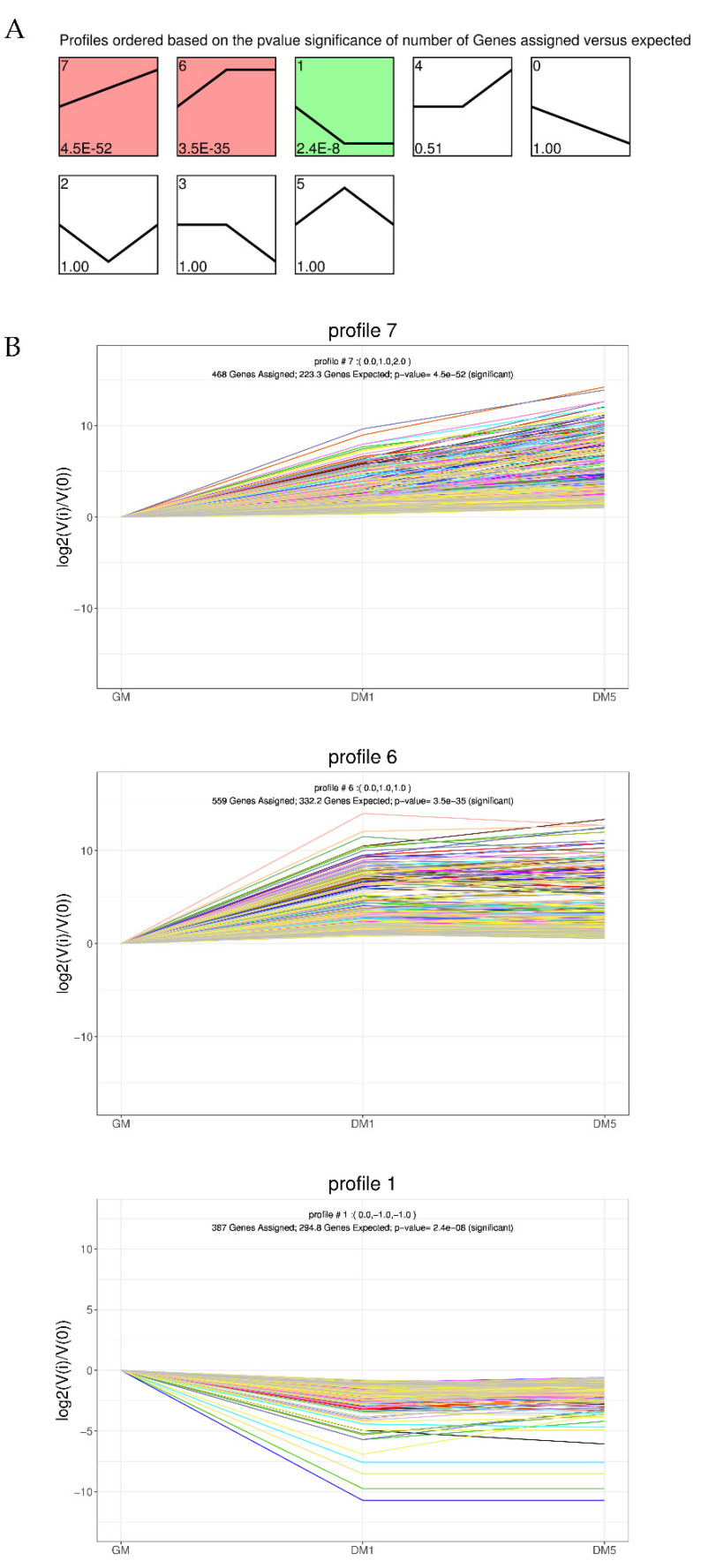
STEM analysis of the DEGs’ expression profiles. (**A**) Trend analysis of the different expression genes; color intensity denotes enrichment. (**B**) Three significant clusters of DEG profiles across all three stages.

**Figure 5 animals-12-01048-f005:**
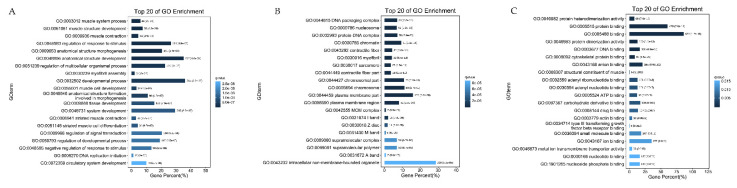
GO enrichment analysis of DM1 vs. DM5. (**A**) The top 20 significance terms of Biological Process in DM1 vs. DM5; (**B**) the top 20 significance terms of Cellular Component in DM1 vs. DM5; (**C**) the top 20 significance terms of Molecular Function in DM1 vs. DM5.

**Figure 6 animals-12-01048-f006:**
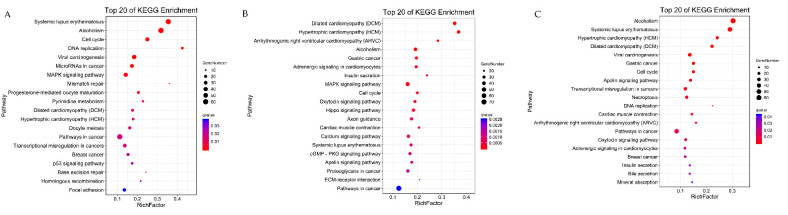
Bubble plot of the top 20 significance pathways of KEGG enrichment. (**A**) The top 20 significance pathways in GM vs. DM1; (**B**) the top 20 significance pathways in GM vs. DM5; (**C**) the top 20 significance pathways in DM1 vs. DM5. The color of the circle represents the q value, which is adjusted to the *p*-value by multiple hypothesis testing. The size of the circle indicates the number of annotated differentially expressed genes.

**Figure 7 animals-12-01048-f007:**
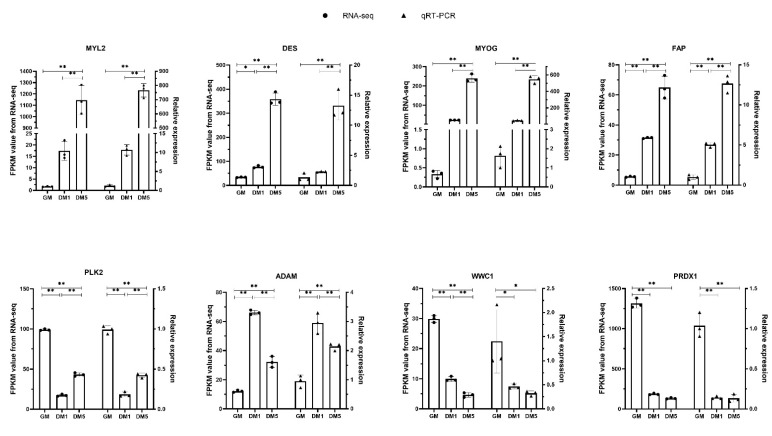
The validation of eight DEGs by qRT-PCR. “*” was considered a significant difference (*p* < 0.05); “**” was considered a highly significant difference (*p* < 0.01).

**Table 1 animals-12-01048-t001:** Primer sequences for qRT-PCR validation.

Gene	Primer Sequence(5′-3′)	Tm/°C	Product Size/bp
*MYL2*	F: GGAGTGCTCAAGGCTGATTATG R: GGCGAACATCTGCTCAATCTC	63.3	87
*DES*	F: GCCGGATCAACCTCCCTATC R: ATGGACCTCAGAACCCCTCT	63.3	83
*MYOG*	F: GGACCCTACAGATGCCCACAA R: TTGGTATGGTTTCATCTGGG	59.0	101
*FAP*	F: CGACCTTACAAACGGGGAGT R: TTTACTCCCAACAGGCGACC	65.0	85
*PLK2*	F: TTCAGTGGGTCACGAAGTGG R: TTGTTCAGGGGCATCTGTGG	55.7	191
*ADAM*	F: TCCAGTTGCACAAAGGTGGT R: GGCAGTGAATCTGGTCTGGT	55.0	135
*WWC1*	F: CGGATGCTGTGTCTGCTCTGTT R: GGTCTGCGTGCTGCTCCTTT	63.3	80
*PRDX1*	F: AGCCTAGCTGACTACAAAGGAA R: GTGTTGATCCATGCCAGGT	59.0	182
*GAPDH*	F: GCAAGTTCCACGGCACAG R: GGTTCACGCCCATCACAA	59.0	249

**Table 2 animals-12-01048-t002:** Summary of the RNA-Seq data in MuSCs.

Samples	Raw Reads	Clean Reads (%)	Unique-Mapped Reads	GC Content	Q30 Value
GM-1	79,898,070	79,694,066 (99.74%)	77,518,765 (90.59%)	49.96%	93.74%
GM-2	82,449,876	82,218,204 (99.72%)	70,235,626 (90.44%)	50.41%	94.15%
GM-3	78,770,454	78,549,636 (99.72%)	70,114,545 (89.61%)	49.92%	93.90%
DM1-1	85,834,132	85,610,188 (99.74%)	71,993,906 (90.73%)	50.31%	94.18%
DM1-2	77,919,438	77,705,900 (99.73%)	71,317,809 (90.66%)	50.49%	94.27%
DM1-3	78,474,104	78,271,608 (99.74%)	72,775,979 (91.04%)	50.82%	94.06%
DM5-1	79,580,920	79,392,054 (99.76%)	72,103,450 (90.51%)	49.86%	94.20%
DM5-2	78,868,496	78,701,026 (99.79%)	74,380,716 (90.51%)	51.30%	94.01%
DM5-3	80,154,802	79,979,248 (99.78%)	71,321,563 (90.84%)	49.57%	93.85%

## Data Availability

The results from data analyses performed in this study are included in this article and its tables. The raw sequencing data are available through the NCBI data accession number PRJNA779184.

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
