# Peer review of "Profiling and Functional Analysis of mRNAs during Skeletal Muscle Differentiation in Goats"

_animals, 2022, doi:10.3390/ani12081048_

Round 1

Reviewer 1 Report

Evaluation

An attempt was made to identify the expression of genes responsible for myogenesis in goats.

The mRNA-Seq studies were conducted on goat muscle satellite cells (MuSCs) cultured under proliferation (GM) and differentiation (DM1/DM5). 19,871 goat genes were expressed, including 198 new transcriptions.

Transcriptomic changes in muscle satellite cells (MuSCs) cultured under proliferation and differentiation conditions were analysed and several candidates identified.

General comments:

Samples of the longest muscle were isolated from how many individuals and which sex were then used in a tissue culture.

Standard molecular methods were used: RNA Extraction and Sequencing, Differentially Expressed Gene Analysis, Expression Trend Analysis, Gene Ontology and KEGG Pathway Analysis, Validation of Sequencing Results by qRT-PCR

A one-way analysis of variance was used to verify this type of data.

The results were presented in a transparent and comprehensible manner.

Chapter Discussion

In the Discussion chapter, the authors relate their results sufficiently to the results of other authors. The main factors responsible for myogenesis are well described.

The great value of this work is the phenotypic expression characteristic of MuSCs in all phases of muscle development, i. e. proliferation and myogenic differentiation.

The work is scientifically very valuable and I think it should be published in the journal Animals.

Evaluation

An attempt was made to identify the expression of genes responsible for myogenesis in goats.

The mRNA-Seq studies were conducted on goat muscle satellite cells (MuSCs) cultured under proliferation (GM) and differentiation (DM1/DM5). 19,871 goat genes were expressed, including 198 new transcriptions.

Transcriptomic changes in muscle satellite cells (MuSCs) cultured under proliferation and differentiation conditions were analysed and several candidates identified.

General comments:

Samples of the longest muscle were isolated from how many individuals and which sex were then used in a tissue culture.

Standard molecular methods were used: RNA Extraction and Sequencing, Differentially Expressed Gene Analysis, Expression Trend Analysis, Gene Ontology and KEGG Pathway Analysis, Validation of Sequencing Results by qRT-PCR

A one-way analysis of variance was used to verify this type of data.

The results were presented in a transparent and comprehensible manner.

Chapter Discussion

In the Discussion chapter, the authors relate their results sufficiently to the results of other authors. The main factors responsible for myogenesis are well described.

The great value of this work is the phenotypic expression characteristic of MuSCs in all phases of muscle development, i. e. proliferation and myogenic differentiation.

Author Response

Thank you for your comments concerning our manuscript. These comments are all valuable and very helpful for revising and improving our paper. We have studied comments carefully and have made corrections which we hope meet with approval. The responds to the reviewer’s comments are as follows:

General comments:

Samples of the longest muscle were isolated from how many individuals and which sex were then used in a tissue culture.

Reply: Thanks. Chengdu Ma goats were used in this study. Longissimus dorsi muscle samples were taken from a female fetus (at 120 days of gestation). These details have been included in the updated manuscript.

Standard molecular methods were used: RNA Extraction and Sequencing, Differentially Expressed Gene Analysis, Expression Trend Analysis, Gene Ontology and KEGG Pathway Analysis, Validation of Sequencing Results by qRT-PCR.

A one-way analysis of variance was used to verify this type of data.

The results were presented in a transparent and comprehensible manner.

Reply: Thanks for your positive comments.

Chapter Discussion

In the Discussion chapter, the authors relate their results sufficiently to the results of other authors. The main factors responsible for myogenesis are well described.

The great value of this work is the phenotypic expression characteristic of MuSCs in all phases of muscle development, i. e. proliferation and myogenic differentiation.

The work is scientifically very valuable and I think it should be published in the journal Animals.

Reply: Thanks for your positive comments.

Reviewer 2 Report

Dear authors, your work inspires respect for its quality and clarity of presentation. I have only a couple of recommendations and one remark mention.

First, the discussion lacks a comparison of the findings with those of other animals or human. Secondly, it would be appropriate to present in more detail the fundamental and applied significance of your results. Thirdly,  the limitations of the methods of analysis used (first of all an enrichment analyses) should be pointed out at least in general terms,
A quick note: stating that Domestic goats are primarily kept for meat production raises many questions (do you mean total numbers of animals, total incomes, meat vs milk and fiebre breeds number? etc) and requieres a data source. I'd rephrase the statement saying, for example, that "goat meat production is highly important for many economics. especially in.."  

Thank you for your interesting work.

Author Response

Thank you for your comments concerning our manuscript. These comments are all valuable and very helpful for revising and improving our paper. We have studied comments carefully and have made corrections which we hope meet with approval. The responds to the reviewer’s comments are as follows:

Comments and Suggestions for Authors

Dear authors, your work inspires respect for its quality and clarity of presentation. I have only a couple of recommendations and one remark mention.

Reply: Thanks for your positive comments.

First, the discussion lacks a comparison of the findings with those of other animals or human. Secondly, it would be appropriate to present in more detail the fundamental and applied significance of your results. Thirdly, the limitations of the methods of analysis used (first of all an enrichment analyses) should be pointed out at least in general terms.

Reply: Thanks for your comments. We have added these information according to your suggestion. More details have been included in the updated manuscript.

A quick note: stating that Domestic goats are primarily kept for meat production raises many questions (do you mean total numbers of animals, total incomes, meat vs milk and fiebre breeds number? etc) and requieres a data source. I'd rephrase the statement saying, for example, that "goat meat production is highly important for many economics. especially in.." 

Reply: Thanks for your comments. We rephrased this sentence. Please see line 71-73: “Domestic goats (Capra hircus) are one of the world's earliest domesticated livestock species, with meat, milk, hair, and skins used all over the world. Goat meat production is crucial for a variety of industries, particularly in the livestock business of developing countries.”

Thank you for your interesting work.

Reply: Thanks for your positive comments.

Reviewer 3 Report

Review Report Animals-MDPI

Brief summary

The manuscript focused on profiling and functional analysis of mRNAs during skeletal muscle differentiation in goat. The study wants to evaluate DEGs of MuSCs through NGS (Illumina Hi-seq Sequencing) and qRT-PCR. The methodology and statistical analysis is very robust and experimental design is correct. Anyway, I have some doubt about the applicability of the study, so I expect some clarifications. The paper is well written and readable.

Broad comments

The manuscript focused on profiling and functional analysis of mRNAs during skeletal muscle differentiation in goat. The study wants to evaluate DEG of MuSCs through NGS and qRT-PCR. The methodology and statistical analysis is very robust and experimental design is correct. Graphics and figures are very well represented in support of data and results reported in the main text.

Anyway, I have some doubts about the applicability of the study. The different level of expression of investigated genetic patterns should be reasonable and expectable in the different stages of skeletal myogenesis, in particular in cellular growth and differentiation so I’m waiting for some clarifications about the expected and obtained results.

In my opinion, the introduction has to be implemented also with some recent lacking references, better framing the topic and the background, clarifying the aim and the applicability of the work.

Line 70-71: this statement has to be contextualized and it is not definitely real, because, to the best of my knowledge, domestic goats (Capra hircus) is also characterized by milk attitude.

Have you available information about goats breed? Did you evaluate the possibility to conduct a larger and not preliminary study aimed at highlighting DEGs inter and intra-breed?

Line 296: probably you wanted to write “further” (?)

English is correct; the paper is well written and readable for scientific community.

All the acronymous along the text have to be defined.

References

Please check the reported references also in accordance with the format required by “Animals-MDPI”

Author Response

Thank you for your comments concerning our manuscript. These comments are all valuable and very helpful for revising and improving our paper. We have studied comments carefully and have made corrections which we hope meet with approval. The responds to the reviewer’s comments are as follows:

Comments and Suggestions for Authors

Review Report Animals-MDPI

Brief summary

The manuscript focused on profiling and functional analysis of mRNAs during skeletal muscle differentiation in goat. The study wants to evaluate DEGs of MuSCs through NGS (Illumina Hi-seq Sequencing) and qRT-PCR. The methodology and statistical analysis is very robust and experimental design is correct. Anyway, I have some doubt about the applicability of the study, so I expect some clarifications. The paper is well written and readable.

Reply: Thanks for your positive comments.

Broad comments

The manuscript focused on profiling and functional analysis of mRNAs during skeletal muscle differentiation in goat. The study wants to evaluate DEG of MuSCs through NGS and qRT-PCR. The methodology and statistical analysis is very robust and experimental design is correct. Graphics and figures are very well represented in support of data and results reported in the main text.

Reply: Thanks for your positive comments.

Anyway, I have some doubts about the applicability of the study. The different level of expression of investigated genetic patterns should be reasonable and expectable in the different stages of skeletal myogenesis, in particular in cellular growth and differentiation so I’m waiting for some clarifications about the expected and obtained results.

In my opinion, the introduction has to be implemented also with some recent lacking references, better framing the topic and the background, clarifying the aim and the applicability of the work.

Reply: Thanks for your comments. We rephrased this section according to your suggestion.

Line 70-71: this statement has to be contextualized and it is not definitely real, because, to the best of my knowledge, domestic goats (Capra hircus) is also characterized by milk attitude.

Reply: Thanks for your comment. We rephrased this sentence. Please see line 71-73: “Domestic goats (Capra hircus) are one of the world's earliest domesticated livestock species, with meat, milk, hair, and skins used all over the world. Goat meat production is crucial for a variety of industries, particularly in the livestock business of developing countries.”

Have you available information about goats breed? Did you evaluate the possibility to conduct a larger and not preliminary study aimed at highlighting DEGs inter and intra-breed?

Reply: The goat breed used in this study is Chengdu Ma goat, a local goat breed. Thanks for your suggestion, it's a good idea. We will consider highlighting DEGs inter and intra-breed in future studies.

Line 296: probably you wanted to write “further” (?)

Reply: Yes. We have revised it.

English is correct; the paper is well written and readable for scientific community.

Reply: Thanks for your positive comments.

All the acronymous along the text have to be defined.

Reply: Thanks. We have added this information according to your suggestion.

References

Please check the reported references also in accordance with the format required by “Animals-MDPI”

Reply: Thanks. We have checked the format of the references according to your comment, and confirmed that the format is in line with the requirements.

Round 2

Reviewer 3 Report

Dear Authors,

the quality of the manuscript was improved as well as the concern/topic contextualization. I suggest to check some english language and style. Furthermore Table 2 lacks of the legend, with abbreviation and acronymous explaination.

Best regards.

This manuscript is a resubmission of an earlier submission. The following is a list of the peer review reports and author responses from that submission.